# In Vitro Biocompatibility and Degradation Analysis of Mass-Produced Collagen Fibers

**DOI:** 10.3390/polym14102100

**Published:** 2022-05-21

**Authors:** Kiran M. Ali, Yihan Huang, Alaowei Y. Amanah, Nasif Mahmood, Taylor C. Suh, Jessica M. Gluck

**Affiliations:** Department of Textile Engineering, Chemistry and Science, Wilson College of Textiles, NC State University, Raleigh, NC 27695, USA; kmmumtaz@ncsu.edu (K.M.A.); yhuang38@ncsu.edu (Y.H.); aya8@duke.edu (A.Y.A.); nmahmoo3@ncsu.edu (N.M.); tacook3@ncsu.edu (T.C.S.)

**Keywords:** collagen, tissue engineering, biomaterials, biocompatibility, scaffolds

## Abstract

Automation and mass-production are two of the many limitations in the tissue engineering industry. Textile fabrication methods such as electrospinning are used extensively in this field because of the resemblance of the extracellular matrix to the fiber structure. However, electrospinning has many limitations, including the ability to mass-produce, automate, and reproduce products. For this reason, this study evaluates the potential use of a traditional textile method such as spinning. Apart from mass production, these methods are also easy, efficient, and cost-effective. This study uses bovine-derived collagen fibers to create yarns using the traditional ring spinning method. The collagen yarns are proven to be biocompatible. Enzymatic biodegradability was also confirmed for its potential use in vivo. The results of this study prove the safety and efficacy of the material and the fabrication method. The material encourages higher cell proliferation and migration compared to tissue culture-treated plastic plates. The process is not only simple but is also streamlined and replicable, resulting in standardized products that can be reproduced.

## 1. Introduction

Disease, injury, and trauma can damage and degenerate tissues in the human body that can be treated via the repair, replacement, or regeneration of the tissue [1]. Often, the tissues are either taken from the patient’s own body (autograft) or from a donor (allograft). An example of an autograft is a bone graft, which is considered the “golden standard” treatment for spinal surgery [2]. Bone, tendons, cartilage, skin, heart valves, and veins are some of the tissues that can be treated using these methods [3]. This type of treatment has been revolutionary and has saved numerous lives. However, the treatment has many drawbacks, including cost and a sometimes difficult recovery, whereas a transplant may result in an immune response or even organ rejection [4].

For this purpose, tissue engineering is beneficial, as tissues are regenerated instead of being replaced. It involves the expansion of cells from a patient’s biopsy with the help of ex vivo cell culture that function together to create tissues and organs. Tissue regeneration is carried out with the help of biological support in the form of engineered scaffolds, which restore or improve cell function [1]. Scaffolds play an important role in the manipulation of cell function and guidance toward new tissue development [5]. 

Various biomaterials have been used for the fabrication of scaffolds for regenerative medicine. Among them, collagen is one of the most prominent proteins due to its excellent biocompatibility [6]. Its high flexibility, optimal mechanical strength, and ability to absorb fluids, makes it an ideal biopolymer. Collagen is also one of the most prominent proteins found in the extracellular matrix (ECM) of humans and is expressed in tissues throughout the human body such as tendons, ligaments, bones, the dermis, dentins, and blood vessels [7,8,9]. Additionally, bovine and porcine collagens have shown degradable properties and have been used for the fabrication of biodegradable medical devices/scaffolds [10,11], such as drug delivery systems [12], and in bone tissue repair [13]. In a clinical study by Charriere et al. [14], the immunological response to bovine collagen implants was evaluated. Out of 705 participants, only 2.3% of the patients showed an adverse reaction to the collagen implant. Another study by Keefe et al. [15] also showed that the adverse effects were only observed in 1–2% of the treated patients, which were absolved when the material was resorbed by the hosts. For this reason, collagen polymers, especially porcine and bovine collagen polymers, are widely used in regenerative medicine.

There are multiple fabrication methods for the creation of scaffolds, such as solvent casting/particle leaching, thermally induced phase separation (TIPS), and three-dimensional (3D) printing [16]. Although these techniques have been very advantageous for the creation of scaffolds, they all have disadvantages such as a lack of reproducibility, low production efficiency, and reduced mechanical strength [16,17]. The most common problems with these methods are reproducibility, repeatability, and standardization.

Textile fabrication methods, such as ring spinning, knitting, weaving, and braiding, have been used for centuries. Most of these fabrication techniques, including for yarn and fabric production, have now been automated and standardized, resulting in efficiently reproducible products. Using these methods to produce scaffolds will provide us with the ability to mass-produce products that are identical and that can be created in a standardized manner [18]. 

Traditional textile manufacturing methods, including knitting, weaving, and electrospinning, have been used for scaffold fabrication. There is a resemblance between the structure of native ECM and textile fibers [18]. The yarn-like structure of ECM resembles the fibrous morphology of textile materials, which has been proven to be beneficial for cellular migration and proliferation [19,20]. The fibrous architecture of ECM helps to provide support to the cell by creating a mesh of collagen, elastin, and other proteins. It is because of the fibrous structure of ECM that the cells are able to form a communication/signaling network [20]. Additionally, textile methods also create 3D structures, which are desirable for cellular scaffolds in terms of cell proliferation and migration [21]. 

Different methods are available for spinning yarns, including the ring spinning, rotor spinning, wrap-spinning, and core-spinning methods [22] (Table 1). Among these, ring spinning is the most popular method for creating yarns using staple fibers (short fibers ranging from 10–500 mm in length). The spun yarns can be used to create knit, woven, or braided structures for tissue engineering applications. 

Weaving provides strength to the yarn. A study by Gilmore et al. [26] used round and grooved cross-sectional fibers to create plain weave and satin weave scaffolds. The results suggested that the fabric structure has a significant influence on the resultant properties, especially on the permeability of the scaffold. Similarly, knitting provides elasticity and strength to fibrous scaffolds. A study by Lieshout et al. [27] shows a comparison of knitted and electrospun scaffolds for aortic valves. Human myofibroblasts were cultured on both scaffolds over the course of 23 days. The results compared tissue formation, which was evaluated via confocal laser scanning microscopy. The study showed that the electrospun scaffold tore within 6 h, while the knit scaffold remained intact. Another study by Zhang et al. [28] explored the idea of using a circular knit machine to produce a small-caliber vascular graft. The scaffold had excellent mechanical properties (bursting strength, suture retention strength, and compliance) that were comparable to the coronary artery under normotensive pressure.

This study evaluates an effort to mass-produce collagen fiber scaffolds that can be fabricated via traditional textile processes (Figure 1). In this study, collagen yarns are produced using the ring-spinning method, which is a process of creating yarns by twisting fibers together and winding them on a bobbin. The method comprises four processes: carding, drawing, roving, and ring spinning (Figure 2). The purpose of carding and drawing is to clean and align the fibers. After drawing, the fibers are passed to a roving machine, which fuses the fibers together through twisting, creating a roving. A roving is a sliver of fibers that is larger in diameter than yarn. The roving is then passed to a ring spinning machine, which provides the fibers with more twist, reducing the diameter to convert them into a yarn [22]. As a result, yarns are stronger when they are fabricated via ring spinning than other methods, such as rotor spinning. Ring spinning also provides a high production rate and can be used for any type of fiber [23].

After the yarns were fabricated, they were subjected to biocompatibility and biodegradation analysis to evaluate the perspective use of this technique in the future to mass-produce scaffolds. The biocompatibility analysis provides knowledge about the safety and efficacy of the fabricated when encountering cells. This is crucial for the application of this process in tissue engineering products, especially for dermal [29], cardiovascular [28], tendon [30], and ligament [18] applications. 

## 2. Materials and Methods

Collagen fibers were provided by the Kaneka Corporation. The fibers were processed through a card chute feed system (Rieter Card C4) followed by two rounds of drawing (Rieter RSB851). The aligned fibers were then converted to a roving with a roving machine (Rieter Fly F4/1) and finally spun into yarn on a ring spinning machine (Rieter G5/2) (Figure 2). The yarns had a fineness of 20 Ne, which is the thickness or coarseness of a yarn, and were measured according to the English Cotton Count (Ne) method, which is defined as “the number of 840-yard length per pound” [23]. Tensile strength and ultrastructure characterization were carried out via scanning electron microscopy (SEM), and an in vitro biocompatibility analysis was performed on the resulting yarns (Figure 1).

### 2.1. Degradation Study

A degradation study was conducted to check the ability of the material to degrade when implanted in the body or when used as an in vivo application. Collagenase is normally found in almost all mammalian tissues, including pig pancreas, beef pancreas, and human tissues. It helps in wound healing by moving the keratinocytes over the collagen-rich dermis during re-epithelialization.

A collagenase solution was prepared according to the protocol by Alberti et al. [31]. Collagenase isolated from *Clostridium histolyticum* cleaves the bonds between neutral amino acids and glycine, which is found with a high frequency in collagen. This type of degradation is a simplistic method to predict the in vivo degradation profile of our ring-spun collagen yarns. First, a collagen yarn sample was dried, weighed, and placed in 12-well plates. In a separate glassware, 0.5 mL of 0.1 M TRIS-HCl and 0.005 M of CaCl_2_ were mixed with 2 mg/mL of collagenase (Type I, powder, Gibco, >125 units/mg). A 2 mL amount of the prepared solution was added to each well such that each sample was immersed in the solution, and the samples were placed in an incubator at 37 °C for a total of 8 weeks. Their weight, tensile strength, and morphology were monitored biweekly. The ultrastructure and morphology of the yarns were assessed using SEM to evaluate the degradation pattern in the collagen yarns. 

#### 2.1.1. Weight Change

A total of 6 samples were prepared, all of which were 20 cm each in length. It was determined that the length would be longer to increase the accuracy. The samples were weighed and placed in a 12-well plate and immersed in collagenase solution for the entire length of the study. The samples were then placed in the incubator at 37 °C. The solution was maintained at a pH of 7–7.4 throughout the study. The samples were dried every 14 days by aspirating the solution and drying in the desiccator for 24 h. After drying, their weight was measured, and the samples were placed in the original well plates again and submerged in collagenase solution at 37 °C. Average mass loss was calculated with the following formula:$$\mathrm{Mass}\;\mathrm{Loss}=\frac{\mathrm{Original}\;\mathrm{Mass}-\mathrm{Final}\;\mathrm{Mass}}{\mathrm{Original}\;\mathrm{Mass}}$$

#### 2.1.2. Tensile Test

A total of 24 samples were prepared, each of which were 20 cm in length, and the samples were placed in 12-well plates. Collagenase solution was added to each well, and the samples were stored in the incubator at 37 °C. The pH of the samples was maintained at 7–7.4. Every 14 days, 6 samples were dried and were placed in the desiccator for 24 h. When they had dried completely, the samples were mounted on cardboard (as shown in Figure 3a). An MTS Criterion 43 Tensile tester (MTS Systems, Eden Prairie, MN USA) was used for the tensile test, and a 50 N load cell, a gauge length of 1 cm, and a crosshead speed of 10 mm/min were maintained for the tensile test.

#### 2.1.3. Scanning Electron Microscopy

The samples were dried and mounted on stubs using double-sided carbon tape. They were sputter-coated with gold/palladium using a SC7620 Mini Sputter Coater (Quorum Technologies, East Sussex, UK) for 45 s, resulting in a 10 nm coating. Samples were imaged with a Phenom G1 desktop SEM (Phenom, ThermoFisher, Eindhoven, The Netherlands). A total of 9 samples were analyzed per time point, and 4 images were taken from each sample. The morphology of the yarns was observed at biweekly intervals. The images were analyzed for surface and bulk degradation. 

### 2.2. Biocompatibility

#### 2.2.1. Sample Preparation

The collagen yarns were sterilized by immersing in 70% ethanol for 20 min. The ethanol was then aspirated followed by washing with phosphate-buffered solution (PBS, Hyclone, Cytiva, Long, UT, USA).

#### 2.2.2. Cell Culture

Mouse fibroblast cells (NIH-3T3) were used for all of the biocompatibility studies. Cells were maintained in Dulbecco’s Modified Eagle Medium (DMEM, Invitrogen, Waltham, MA, USA) supplemented with 10% fetal bovine serum (FBS, Atlas Biologicals, Fort Collins, CO, USA) and 1% penicillin-streptomycin (10,000 U/mL, Gibco, Invitrogen, Waltham, MA, USA). Cells were seeded onto the collagen yarns at a passage of 9–11 and at a density of 2 × 10^6^ cells/sample. Cell-only controls were seeded at a density of 0.5 × 10^6^ cells/well. The cell-seeded scaffolds were then maintained in culture for 7 days. The medium was changed every 72 h. Instead of aspirating the medium, the samples were transferred into another well, and more medium was slowly added over the seeded samples. This was to avoid cells detaching from the surface of the collagen yarn surface. Cell proliferation, functioning, and morphology were checked.

The biocompatibility was then evaluated via a Live/Dead™ assay (Invitrogen, Waltham, MA, USA), alamarBlue proliferation assay (Invitrogen, Waltham, MA, USA), immunocytochemistry, and SEM analysis of the seeded scaffolds. The live/dead assay and alamarBlue were evaluated on days 1, 3, 5, and 7, whereas phalloidin (F-actin) staining and SEM were carried out on days 1 and 7.

#### 2.2.3. Cell Proliferation

The live/dead assay was carried out using an Invitrogen Live/Dead^TM^ Cell Imaging kit following the manufacturer’s instructions. The solution from the kit was diluted to a ratio of 1:1 with PBS. After mixing the solution, 250 µL of the prepared working solution was added to each scaffold sample and placed in the incubator at 37 °C and 10% CO_2_ for 20 min. The samples were then observed under a fluorescent microscope (EVOS™FL Auto 2 Imaging System, ThermoFisher, Waltham, MA, USA) to check for green (live) and red (dead) fluorescence. The kit contains calcein; AM, a permanent dye that is an alive cell indicator (ex/em 488 nm/515 nm); and BOBO-3 Iodide, which is a dead cell indicator (ex/em 570 nm/602 nm). This test was performed on cell-seeding days 1, 3, 5, and 7. 

The alamarBlue assay was also performed to quantify the proliferation of the cells that had been seeded on collagen fiber scaffolds. This assay provides quantitative data on cell metabolism (proliferation) by measuring the changes in fluorescence generated by resazurin, which reduces to resorufin in response to chemical reductions due to cell growth. The dye changes its color and fluorescence from blue/non-fluorescent (oxidized) to red/fluorescent (unoxidized). 

The cells were seeded on the yarn for 7 days. The alamarBlue assay was carried out on days 1, 3, 5, and 7 with the positive control cells only and with the negative controls of the yarn in media only (no cells). The samples were incubated with alamarBlue Reagent (Fisher Scientific, Waltham, MA, USA) for 1 h. Later, the samples were read with a plate reader (Synergy HT, BioTek, Santa Clara, CA, USA) set to 540/25 λ excitation, 590/35 λ emission, and maintained at 37 °C.

#### 2.2.4. Phalloidin Staining

Phalloidin staining was carried out to label and identify F-actin (cytoskeleton) expression in the cells to observe the cellular morphology. On day 7, the samples were fixed with 4% paraformaldehyde for 20 min and washed with PBS. The samples were then permeabilized with 0.1% TritonX-100 for 30 min and 0.1% Tween-20 for 15 min. Blocking buffer was prepared with 0.1%Tween-20, 2% bovine serum albumin, and 2% goat serum. Phalloidin (Invitrogen™ ActinGreen™ 488 ReadyProbes™ Reagent) was used in the concentration of 2 drops/mL of blocking buffer and incubated for 1 h. After incubation, the samples were washed with PBS and stained with Hoechst (Invitrogen™ Hoechst 33342, Eugene, OR, USA) (1:1000 concentration Hoechst: PBS) for 5 min. The samples were then imaged using fluorescence microscopy (EVOS™ FL Auto 2 imaging system, ThermoFisher, Waltham, MA, USA).

#### 2.2.5. Ultrastructure Analysis (Scanning Electron Microscopy)

Collagen fiber yarns seeded with NIH 3T3 fibroblasts were fixed in buffered formalin. The samples were washed 3 × 10 min with 0.15 M sodium phosphate buffer and at a pH of 7.4 followed by post-fixation in 1% osmium tetroxide/0.15 M sodium phosphate buffer at a pH of 7.4 for 1 h. After washing with deionized water (3 × 10 min), the samples were treated with 1% tannic acid in water for 30 min. The scaffolds were washed in deionized water and dehydrated through an increasing ethanol series (30%, 50%, 75%, 90%, 100%, 100%, and 100%—15 min each). The samples were transferred in 100% ethanol to a Samdri-795 critical point dryer (Tousimis Research Corporation, Rockville, MD, USA) and dried using liquid carbon dioxide as the transitional solvent. The scaffolds were mounted onto 13 mm diameter aluminum stubs with carbon adhesive tabs and were sputter-coated with 8 nm of gold/palladium alloy (60Au: 40Pd) using a Cressington 208HR Sputter Coater (Ted Pella, Inc., Redding, CA, USA). Images were taken using a Zeiss Supra 25 FESEM (Carl Zeiss Microscopy, Jena, Germany) operating at 5 kV or 10 kV using the SE2 detector, 30 μm aperture, and approximate working distances from 15 to 25 mm. The samples were imaged for any cell structures that were growing horizontally or vertically over the fiber and yarn surfaces. 

### 2.3. Statistical Analysis

Data were collected from 3 to 9 samples and expressed as means ± standard deviation. Statistical analysis was performed using the Student’s *t* test, and significance was determined at *p* < 0.0001.

## 3. Results 

This study evaluated the biodegradability and biocompatibility of mass-produced ring spun collagen fiber yarns (Figure 1). The collagenase (Type I) enzyme was used to assess yarn degradation. Mass changes, tensile strength, and the morphology of the yarns were checked at biweekly intervals. The biocompatibility of the yarns was also checked using mouse fibroblast cells. The cell viability, cell metabolic activity, and the morphology of the cytoskeleton was observed for biocompatibility analysis.

### 3.1. Collagen Yarns Degrade in the Presence of Enzymes in an 8-Week Study

During the enzymatic degradation with collagenase Type I, the collagen yarns lost most of their mass and mechanical integrity over 8 weeks. On average, the samples lost 80 ± 6% of their mass over 8 weeks (Figure 3c), which indicates that the material is degradable, and the remnants were dispersed in the solution. The pH of the solution remained the same throughout the study. The degradation of collagen fibers was confirmed via the assessment of the tensile strength and morphology of the samples, as indicated by the losses observed in the measured Young’s modulus (Figure 3b). The modulus loss indicates the degradation of the collagen fibers in the presence of collagenase solution. The samples lost 97% of their tensile strength in 6 weeks. Fiber samples were mechanically unstable by week 6 and therefore could not be tested for tensile strength in week 8.

Gradual degradation was evident in the yarns for eight weeks (Figure 4) when week 0 and week 8 are compared. A polymer undergoes bulk degradation if the diffusion of water/fluids into the polymer is faster than the bonds break [32]. The SEM image analysis of the collagen yarns indicated that the length of the fibers broken down over 8 weeks (Figure 4 marked with arrows), suggesting bulk degradation. On the other hand, the surface of the fiber also changed from smooth to rough and beady (marked with stars), which is a sign of surface degradation in the fibers. 

### 3.2. The Collagen Material Is Biocompatible with Mouse Fibroblast (NIH 3T3) Cell Line

High cell viability on the collagen yarns was observed over 7 days (Figure 5). After only 24 h of cell culture, the cells were observed to form colonies, which was a strong indication that the material provides a heterogeneous environment where the cells are more attracted to some areas, forming clusters and migrating to a suitable area. The cells adhered to the yarn/fiber surface and proliferated along the length (Figure 5c,d).

The collagen yarns support high cellular viability (Figure 5), and the corresponding metabolic activity confirmed the viability (Figure 5e). The metabolic activity results show an 8x increase in metabolic activity from day 1 to day 5, only to reduce after that. The reduction in proliferation may be due to limited space for the cells to proliferate any further. A study by Streitchan et al. [33] has shown that cells slow down or pause their proliferation cycle due to spatial constraints in response to contact inhibition. This was highlighted on day 5, where we observe a reduction in metabolic activity. Cells proliferated more on the collagen yarns compared to standard tissue culture plastic dishes, indicating their superior biocompatibility. 

The morphology of the cells seeded on the collagen yarns is observed to be a bright ellipsoid shape that wraps around the strands of the collagen fibers, as evidenced by f-actin detection (Figure 6a,b). The ultrastructure analysis from the SEM images (Figure 6c–h) highlights that cell growth can be observed along the direction of the collagen fiber. 

## 4. Discussion

This study demonstrates that textile-based mass-production techniques may be used for the fabrication of biocompatible tissue-engineered scaffolds. The yarns in this study were produced using a traditional ring spinning technique facility where general textile yarns are produced. These yarns can be used to produce woven, knit, or braided structures that can be used for tissue engineering applications. For example, Learn et al. [30] used collagen yarns to fabricate a woven scaffold for rotator cuff tendon regeneration, whereas Ruan et al. [34] used a collagen-silk blend to create knit scaffolds for anterior cruciate ligament reconstruction. The data demonstrate that collagen fiber yarns are biocompatible. The biocompatibility of a material is defined as the properties of a material that are safe and effective for cells to proliferate, migrate, and function [35]. The results from our viability and proliferation assays combined with our fluorescent and electron microscopy image analysis show that both the material and the fabrication method support cellular proliferation and maintain their expected cellular morphology (Figure 5 and Figure 6). It is interesting to observe that the cells migrated from the tissue culture-treated plate to the fiber surface, providing compelling evidence of the biocompatibility of the material.

Similarly, the fabrication method demonstrates the ability of the collagen fibers to degrade over time in the presence of the collagenase enzyme. Collagenase is normally present in most mammalian tissues, including those in humans [36]. Collagenase provides a good imitation of the in vivo environment, where enzymatic activity is observed to cleave native fibrillar collagen. In this study, degradation is seen by a reduction in mass and mechanical integrity over time, which signifies that the fabrication methods provide additional mechanical strength without changing the degradable nature of the polymer. Degradation is an important property for implantable biopolymers in situation where the scaffold is only required for mechanical support until the resulting tissue develops in vivo. This prevents the need for surgical procedures to remove the scaffold when it is no longer required [36]. 

Previously, conventional fabrication techniques such as fiber melts, fiber bonding, melt molding, and fiber composite foam have been used, which can provide the necessary morphology. However, these methods have multiple disadvantages, such as a lack of structural stability or the solvent residue being toxic or harmful [17]. On the other hand, traditional textile fabrication methods have many advantages, such as good mechanical stability, a 3D structure, a fibrous nature, and most importantly, the ability to fabricate products that are identical and that can be produced in bulk [18]. 

Collagen is one of the most extensively used materials in tissue engineering due to its abundant presence in the native ECM, where it provides biological and structural integrity to cells [7]. However, when used as a scaffold, the material does not provide mechanical strength, which limits its applications [37]. Collagen is normally used as a blended material or is chemically cross-linked to provide the required strength to the material [37]. However, the use of ring spinning technology solves this problem due to the twisting of fibers by providing excellent mechanical support. 

Additionally, the fibrous structure of textile yarns replicates the thread-like morphology of ECM [18]. Therefore, using collagen in a textile form provides the required support to the cells whiles also providing a medium for networking and signaling [20]. Moreover, the 3D structure of textiles is also beneficial for cell proliferation and migration. Biomimetic fibrous scaffolds are crucial in tissue engineering when the goal is to best recapitulate the native ECM. Creating fibrous scaffolds from collagen has been repeatedly demonstrated to be beneficial for a variety of tissue engineering applications [38], including scaffolds for heart valve repair [39] and cardiac tissue engineering [40], with bone [34,41] and skin tissue engineering applications being the most common examples. 

A study by Xie et al. [42] demonstrated the potential use of wet spun yarn in tissue engineering applications. However, collagen is sensitive to temperature, and therefore, wet spinning requires the yarns to be cross-linked to acquire the required thermal and chemical stability. On the other hand, the ring spinning method is dry and does not require elevated temperatures. Therefore, it is likely to be a more stable form of spinning for collagen fibers. Another common method using for processing collagen for tissue engineering applications includes chemical cross-linking, which is destructive to the natural helical structure of collagen. Thus, we showed that ring-spun collagen yarns represent an exciting new opportunity for the scalable production of biomimetic biomaterials that can be used for a variety of tissue engineering applications. 

Regardless of the excellent results of the in vitro analysis of the fabricated textile yarns, further studies are required to create compact structures that are either woven, knit, or braided. We are currently working on fabricating knit structures for in vitro studies. 

## 5. Conclusions

Collagen fiber yarns are a highly biocompatible material/technique, as they promote cell adhesion and proliferation. The structure of the material encourages cell adhesion and proliferation without requiring any intricate and complex fabrication technique, which not only saves time but also simplifies and streamlines the fabrication process. The process also provides a standardized fabrication technique for creating identical scaffolds. Furthermore, this proof-of-concept study sets the foundation for further analyses of the underlying textile and biological properties of mass-produced collagen fiber scaffolds that support cell behavior and that can be used for further tissue engineering and regenerative medicine applications.

## Figures and Tables

**Figure 1 polymers-14-02100-f001:**
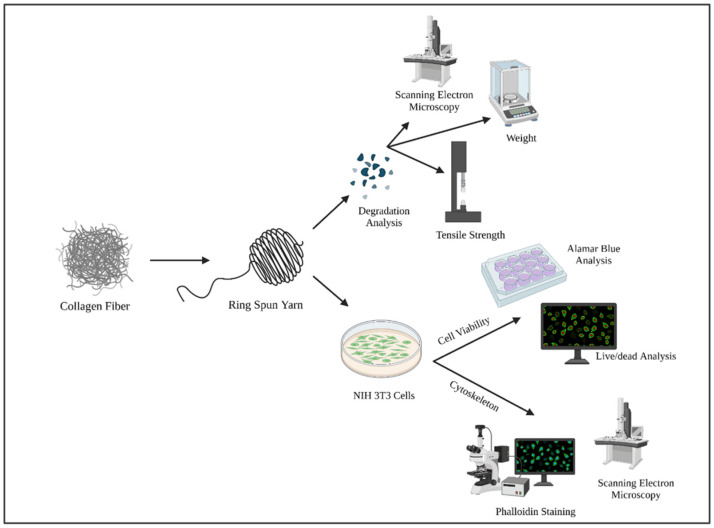
Experimental Design Schematic. Collagen fibers were provided by the Kaneka Corporation and spun to yarns using the traditional ring spinning method. The resulting yarns were analyzed for biodegradability and biocompatibility. The biodegradability of the material was analyzed via weight changes, tensile strength changes, and morphological analysis. The biocompatibility was checked with Alamar blue and live/dead analysis for cell viability, and SEM and phalloidin staining were used for cytoskeleton analysis.

**Figure 2 polymers-14-02100-f002:**
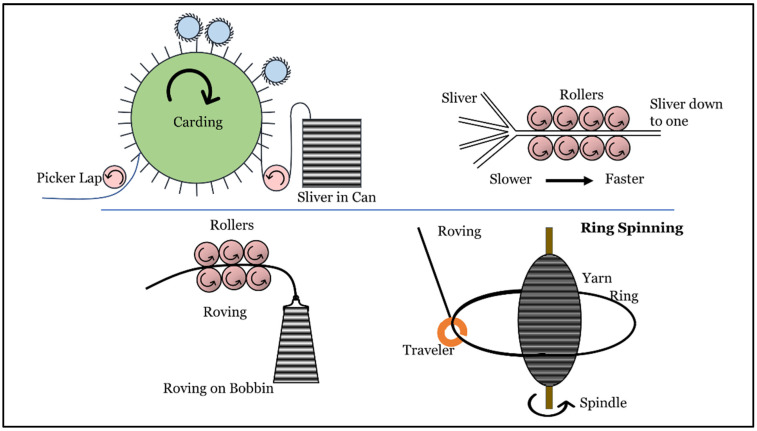
Ring spinning process for collagen yarns. The collagen fibers were provided by the Kaneka Corporation. The fibers were first run through the card and drawer with multiple sets of rollers. The rollers combed the fibers to clean and aligned them before they were gathered to form a sliver in a can. The sliver then passed through a roving frame, where the sliver reduced in diameter to form a roving, which was wound on a bobbin. Finally, the roving bobbins were passed through a ring spinning machine, where their diameters were further reduced via twisting with the help of a traveler and a ring.

**Figure 3 polymers-14-02100-f003:**
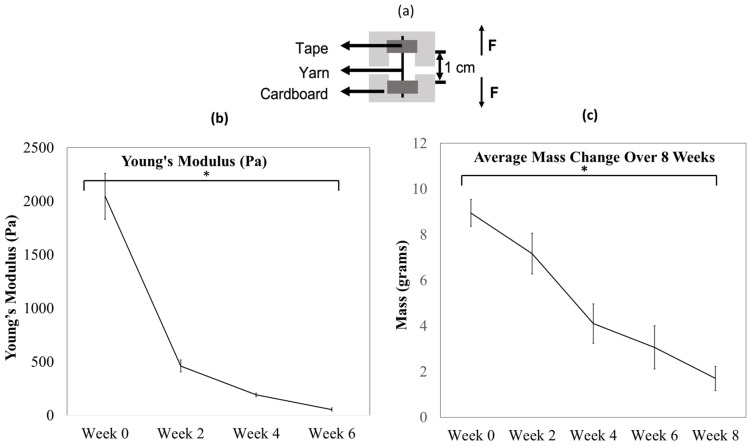
Mechanical analysis over time: Collagen yarns were soaked in a collagenase solution for 8 weeks (*n* = 3 samples per period). Samples were prepared and mounted using a cardboard holder (**a**) for tensile testing. The samples showed a significant loss (97%, * *p* < 0.0001) in strength over 6 weeks. After 6 weeks, the samples lost mechanical integrity and therefore could not be tested further. There was a significant mass reduction * (*p* < 0.0001) from week 0 to week 8. (**b**) Mass change was calculated and recorded over time for the material degradation analysis. In the study, 20/s yarns were soaked in collagenase solution for 8 weeks, (*n* = 6 per period) and an average mass loss of 80 ± 6% was recorded over 8 weeks. Standard deviation is shown by error bars (**c**).

**Figure 4 polymers-14-02100-f004:**
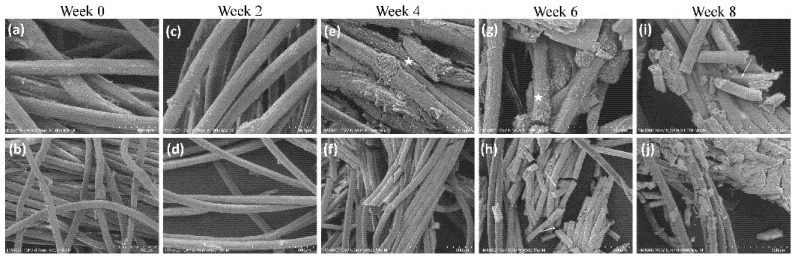
Bulk and surface degradation is observed over 8 weeks. The samples were subjected to collagenase for 8 weeks. The morphology of the fibers was observed via SEM. The fibers had broken down and degraded significantly at week 8 (images **i**,**j**) when compared to week 0 (images **a**,**b**). The surface of the fibers also changed, which shows that there is surface degradation in the fibers (*n* = 6). The top images (**a**,**c**,**e**,**g**,**i**) show a higher magnification (50 um), whereas the bottom images (**b**,**d**,**f**,**h**,**j**) show a lower magnification (100 um). Arrows indicate broken fiber lengths; stars indicate a surface transition from smooth to rough and beady.

**Figure 5 polymers-14-02100-f005:**
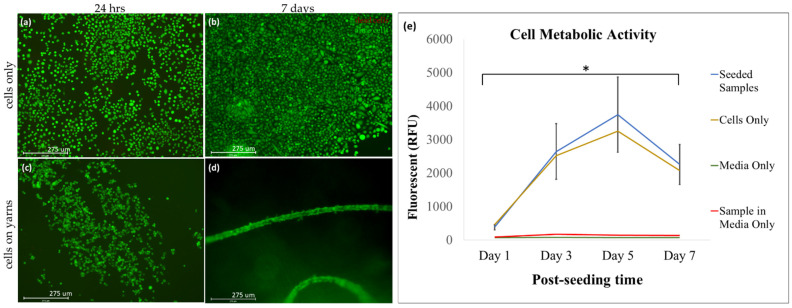
High biocompatibility of collagen yarns is observed. Viability of NIH 3T3 cells (**a**) seeded on collagen yarns. The green signal indicates live cells, and the red signal indicates dead cells. The live/dead assay of the seeded samples at 24 h (**c**) and after 7 days (**d**). The cell-only controls at 24 h and 7 days are also shown (**a**,**b**). We see evidence of the cells adhering and proliferating along the length of the fiber for at least 7 days, indicating the biocompatibility of the collagen yarns (*n* = 10). The results from the live/dead assay were confirmed and quantified with the alamarBlue assay (**e**). We observed an increased in metabolic activity that was correlated to increased proliferation in the cells that were seeded on our collagen yarns (**e**, blue positive control (cells only in yellow) and in the negative controls (medium-only in green and sample in medium in red)). A significant increase in metabolic activity was observed in the cells seeded on collagen yarns from day 1 to day 5 (* *p* < 0.0001). The error bars show the standard deviation (*n* = 9).

**Figure 6 polymers-14-02100-f006:**
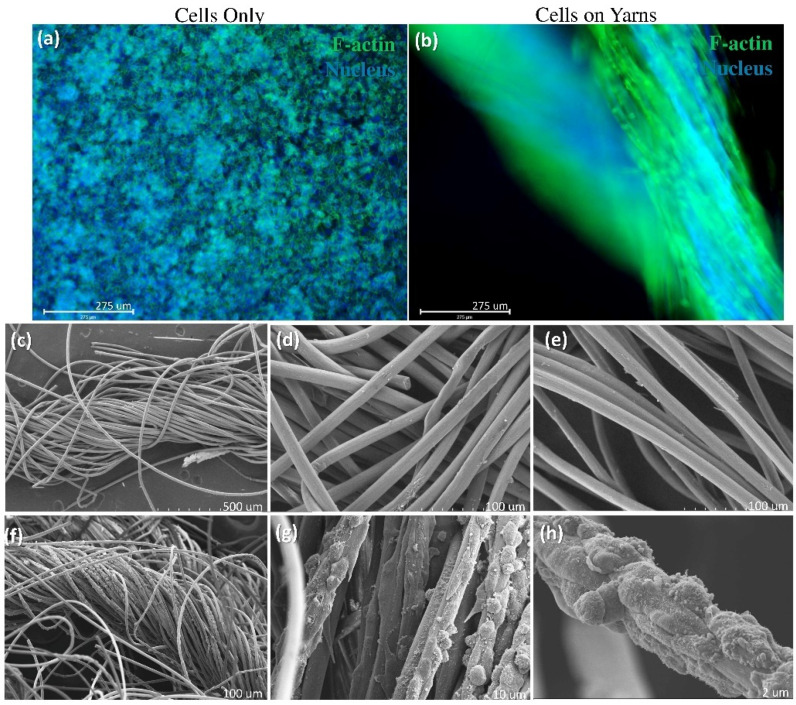
Cells proliferate along the length of the collagen yarns. The samples were stained with phalloidin to identify the cytoskeleton of the cells on the collagen yarns at day 7 (**a**,**b**). F-actin, a cytoskeleton protein, highlights the typical morphology of the cells (**a**) and highlights the adhesion of the cells along the length of the collagen yarns (**b**), scale bar = 275 μm (*n* = 6). Representative SEM images of collagen fibers alone (**c**–**e**) at varying magnifications. NIH 3T3 cells were seeded and maintained on collagen fiber yarns for 7 days (**f**–**h**). Scale bar (**a**,**b**) = 275 μm, (**c**) = 500 μm, (**d**–**f**) = 100 μm, (**g**) = 10 μm, (**h**) = 2 μm (*n* = 3).

**Table 1 polymers-14-02100-t001:** The advantages and disadvantages of different spinning methods.

Method	Advantages	Disadvantages	Ref
Ring Spinning	The yarns that are produced have high strength. This method is applicable to a wide variety of fibers.	The method consumes high amounts of energy, and therefore, production costs are also high.	[22,23]
Rotor Spinning	The production costs of using this method are relatively low.	The resultant strength of the yarns is low.	[22,23]
Wrap-Spinning	The yarns that are produced through this method are highly absorbent. The method has high production efficiency.	The yarns have low strength.	[22,23,24]
Core-Spinning	This method uses two or more fibers, providing excellent properties to the resultant composite yarn.	There are limited applications for this method.	[22,23,25]

## Data Availability

The data presented in this study are available within this publication.

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
