# Peer review of "In Vitro Biocompatibility and Degradation Analysis of Mass-Produced Collagen Fibers"

_polymers, 2022, doi:10.3390/polym14102100_

Round 1
Reviewer 1 Report
The authors used the ring spinning method to produce collagen yarns for tissue engineering applications. This manuscript focused on the biocompatibility and degradation analysis of the yarns. The results are not surprising as collagen is generally believed to be biocompatible and biodegradable. Two major issues must be addressed before publication.
(1) The authors should provide a detailed literature review on the traditional textile methods that can be used for making yarns from collagen fibers, including the advantages and disadvantages of these methods. The authors should pay more attention to introducing textile techniques, which is the merit of this manuscript.
(2) The details of the ring spinning method used in this study must be provided including the make and model of the machine. A photograph or schematic for explaining the mechanical mechanism of the ring spinning of fibers should be provided.
The authors are encouraged to provide a list of target tissue engineering applications for the collagen yarns.
Author Response
We revised our manuscript, “In vitro biocompatibility and degradation analysis of mass-produced collagen fibers” (Manuscript ID polymers-1707554) based on the valuable feedback from the reviewers. By addressing these comments, we believe we have significantly strengthened the manuscript. The revisions are highlighted throughout the text of our revised manuscript and our response to the comments is found below. We thank you for your time and look forward to your decision.
Reviewers' Comments and Our Responses
Reviewer #1
• The authors should provide a detailed literature review on the traditional textile methods that can be used for making yarns from collagen fibers, including the advantages and disadvantages of these methods. The authors should pay more attention to introducing textile techniques, which is the merit of this manuscript.
The literature review has been updated with the addition of traditional textile methods for yarn fabrication (lines 80-99). A table has also been added about the advantages and disadvantages of different spinning methods (Table 1).
• The details of the ring spinning method used in this study must be provided including the make and model of the machine. A photograph or schematic for explaining the mechanical mechanism of the ring spinning of fibers should be provided.
Details of the ring spinning method have been explained in the literature along with an illustration detailing the method (lines 103-109 and Figure 2). Additionally, the make and model of all the machines have been added that were used for the fabrication of yarn (lines 127-130).
• The authors are encouraged to provide a list of target tissue engineering applications for the collagen yarns.
The yarn spinning method, as explained in this manuscript, is used to create yarns which may be used to create tissue engineering scaffolds by weaving, knitting, and braiding. Literature has been added to show previous studies where knitted and woven scaffolds were used for various applications including dermal, cardiovascular, tendon, and ligament tissues. (Lines 338-342, 374-382)
We believe our manuscript now addresses all the reviewer comments. We thank you for your consideration of our manuscript in Polymers, in the special issue “Biomaterials for Tissue Engineering and Regeneration.”

Reviewer 2 Report
Ali's et al. manuscript focuses on the study of the degradation and biocompatibility properties of collagen fibers which are of interests as materials for tissue engineering. Undoubtedly, the topic of the study is urgent and, in general, the manuscript is clearly written. However, the minor revision is required before acceptance.
Comments
- Figures 1, 2 and 4 are blurred. Their quality should be improved.
- Section 3 – Results. Some introductive paragraph explaining the design of study is required.
- Section 3 – Results. The mentioning of the enzyme used for degradation in the first paragraph of section 3.1 is necessary for better understanding of the experimental work.
- Line 134. The citation is failed. Please, correct.
- Lines 220 and 237. The more correct presentation of the result is a value +/- SDV %. Moreover, taking into account that SDV is a whole number, the correct presentation is 80±6 %.
Author Response
We revised our manuscript, “In vitro biocompatibility and degradation analysis of mass-produced collagen fibers” (Manuscript ID polymers-1707554) based on the valuable feedback from the reviewers. By addressing these comments, we believe we have significantly strengthened the manuscript. The revisions are highlighted throughout the text of our revised manuscript and our response to the comments is found below. We thank you for your time and look forward to your decision.
Reviewers' Comments and Our Responses
Reviewer #2
• Figures 1, 2 and 4 are blurred. Their quality should be improved.
Figures 1, 2, and 4 have been improved and are renamed figures 1, 3, and 5 respectively due to the addition of Figure 2.
• Section 3 – Results. Some introductive paragraph explaining the design of study is required.
An introductive paragraph has been added in the results section (lines 253-258).
• Section 3 – Results. The mentioning of the enzyme used for degradation in the first paragraph of section 3.1 is necessary for better understanding of the experimental work.
The name of the enzyme has been added (line 260).
• Line 134. The citation is failed. Please, correct.
Cross-reference has been updated (line 169).
• Lines 220 and 237. The more correct presentation of the result is a value +/- SDV %. Moreover, taking into account that SDV is a whole number, the correct presentation is 80±6 %.
The values in the results have been updated (line 262 and caption for Figure 3, line 278).
We believe our manuscript now addresses all the reviewer comments. We thank you for your consideration of our manuscript in Polymers, in the special issue “Biomaterials for Tissue Engineering and Regeneration.”

Round 2
Reviewer 1 Report
I am convinced that the manuscript is ready for publication.